# Damage Determination in Ceramic Composites Subject to Tensile Fatigue Using Acoustic Emission

**DOI:** 10.3390/ma11122477

**Published:** 2018-12-06

**Authors:** Gregory N. Morscher, Zipeng Han

**Affiliations:** Department of Mechanical Engineering; University of Akron, Akron, OH 44325, USA; zh10@zips.uakron.edu

**Keywords:** acoustic emission, ceramic matrix composites, matrix cracking, fiber breakage

## Abstract

Acoustic emission (AE) has proven to be a very useful technique for determining damage in ceramic matrix composites (CMCs). CMCs rely on various cracking mechanisms which enable non-linear stress–strain behavior with ultimate failure of the composite due to fiber failure. Since these damage mechanisms are all microfracture mechanisms, they emit stress waves ideal for AE monitoring. These are typically plate waves since, for most specimens or applications, one dimension is significantly smaller than the wavelength of the sound waves emitted. By utilizing the information of the sound waveforms captured on multiple channels from individual events, the location and identity of the sources can often be elucidated. The keys to the technique are the use of wide-band frequency sensors, digitization of the waveforms (modal AE), strategic placement of sensors to sort the data and acquire important contents of the waveforms pertinent for identification, and familiarity with the material as to the damage mechanisms occurring at prescribed points of the stress history. The AE information informs the damage progression in a unique way, which adds to the understanding of the process of failure for these composites. The AE methodology was applied to woven SiC fiber-reinforced melt-infiltrated SiC matrix composites tested in fatigue (R = 0.1) at different frequencies. Identification of when and where AE occurred coupled with waveform analysis led to source identification and failure progression. For low frequency fatigue conditions, damage progression leading to failure appeared to be due to fiber failure at or near the peak stress of the stress cycle. For higher frequency fatigue conditions, significantly greater amounts of AE were detected compared to low frequency tests a few hours prior to failure. Damage progression leading to failure included AE detected events which occurred on the unloading part of the fatigue cycle near the valley of the stress cycle. These events were associated with 90 tow longitudinal split and shear cracks presumably due to local compressive stresses associated with mating crack surface interactions during unloading. The local region where these occurred was the eventual failure location and the “valley” events appeared to influence the formation of increased local transverse cracking based on AE.

## 1. Introduction

Ceramic matrix composites (CMCs) comprised of continuous ceramic fibers and ceramic matrices possess high temperature capability and offer higher toughness as compared to monolithic ceramic materials [1]. This is highly desired in applications such as hot-section jet engine components where the additional toughness enables some ease in design [2]. The mechanism for enhanced toughness in properly designed CMCs is the enabling of matrix microfractures due to a weak interface or layer that exists between the fiber and the matrix [3]. For example, when a crack initiates under a tensile load at a local flaw source, it propagates around fibers, debonding along the fiber in this weak interface region, leaving fibers to bridge the crack wake. The strong fibers in the fiber-bridged transverse matrix crack will carry the load shed from the matrix until the applied load is sufficient to fail the fibers in a local region. The result is increased local strain in the matrix crack region and non-linear stress–strain behavior. Ultimate failure then is controlled by the failure of the fibers, which are bridging the matrix cracks at a higher stress.

Acoustic emission (AE) is an excellent technique to monitor the microfracture damage mechanisms in fiber-reinforced CMCs. Since these mechanisms are all microfracture events, part of the energy release that happens when they occur will be in the form of sound (stress) waves transmitted through the material. In reality, there are thousands of microfracture events which occur during the tensile stress–strain response of a CMC. The uniqueness of AE is that it is a passive technique; there is no input, one just listens to what the material tells them. In this sense, one is taking advantage of a “smart sensing” approach to material damage accumulation. Ideally, one would like to be able to monitor when such events occur, what specific type of event occurred and where that event occurred in the volume of interest. AE can or at least has the potential to inform in all three of these areas.

The utility of AE to identify the onset of microfracture damage has been demonstrated in a number of studies. First, AE is probably the best technique to distinguish when initial microfracture damage occurs. This has been demonstrated for CMCs throughout the years [4,5,6]. Initial transverse cracking in SiC fiber-reinforced SiC matrix CMCs is typically small and not detected from non-linearity in the stress–strain curve; therefore, graphical constructs such as offset stress methods to determine “first cracking stress” are insufficient [7]. Since the initial microcracks are typically very small (e.g., 90 tunnel cracks [8]), ultrasonic, X-ray and thermographic non-destructive evaluation techniques are also incapable of detecting this initial low-stress damage in woven SiC/SiC composites. One significant application of this finding is that the stress to cause these initial microcracks correlates with the stress for long-time (1000 h) creep-rupture failure strength [9].

Second, AE has been used effectively in matrix-dominated CMCs [10] during tensile fast-fracture type tests. In these systems, the AE energy of the waveforms is directly related to matrix crack density for both melt-infiltrated [11] and chemically vapor infiltrated matrix SiC fiber-reinforced, SiC matrix composites [12].

Third, AE has been used effectively to locate sources of damage with a resolution of less than ~0.5 mm. This was beneficial in identifying which part of the architecture transverse cracking occurred for a 3D orthogonal SiC/SiC composite [13], where cracks emanated from a notch tip with increasing load [14], and how far cracks propagated around a C-coupon during an interlaminar tension test [15]. More recently, the location analysis coupled with the frequency analysis has enabled the location of transverse crack propagation emanating from a single notch that occurred in the interior (tunnel) plies of a composite and the exterior (surface extending) plies of a composite in a SiC/SiC laminate composite [16].

Fourth, AE has been used to identify sources (mechanisms) based on their waveform characteristics. Already mentioned was high energy, which has been shown to correlate with matrix cracks for tensile tests. In addition, energy content coupled with frequency content has been shown to correlate with interior cracks (high frequency–low energy) versus surface extending cracks (low frequency–high energy) [16,17,18]. This was based on an understanding of plate-wave theory [19,20] where surface cracks being off-center of the midline of a specimen would promote anti-symmetric low frequency flexural waves (which are usually higher in energy for thin plates especially when sensors are mounted on the surface of the specimen) and internal mode 1 cracks being on or near the center line of the composite would promote symmetric extensional waves which always have higher frequencies than flexural waves. This was first demonstrated for the same type of mechanism (transverse cracks at the surface and in the interior) in polymer laminate composites [17,18] and later in SiC/SiC laminate composites [16].

Based on the understanding and methodology described above for waveform-based AE, this study aimed at utilizing these techniques to inform the understanding of damage development for conventional dogbone woven SiC/SiC composite specimens when tested under tensile–tensile fatigue conditions of different frequencies where different AE behavior leading to failure were observed. The differences in AE behavior observed for the different fatigue tests inferred different damage progressions which could be better understood from AE waveform characteristics. This was affirmed by microstructural examination.

## 2. Materials and Methodology

The materials and fatigue tests evaluated in this work were from earlier published work [21,22]. However, this study used the data generated from the earlier works to analyze the AE data more in-depth. The SiC/SiC composites evaluated were of the woven fiber-reinforced slurry-derived melt infiltrated variety. The composites were fabricated by Goodrich Inc. (Santa Fe Springs, CA, USA) prior to their acquisition by United Technologies. The composites were made up of eight plies of 0/90 woven five-harness satin cloth of Hi-Nicalon Type S (Nippon Carbon, Tokyo, Japan) fibers. A boron nitride interphase of approximately 0.5 microns followed by a chemically vapor deposited SiC layer of approximately 2–3 microns was infiltrated to rigidize the structure. This was followed by SiC particle infiltration via an aqueous slurry. After drying the slurry infiltrated preform, the preform was infiltrated with molten Si so that much of the open porosity was filled with Si. It should be noted that these panels had some excess porosity and there was considerable non-uniformity between panels and even within panels. Fatigue tests were performed at 0.01, 0.1 and 1 Hz at an R value of 0.1 at room temperature using an MTS hydraulic universal testing machine (model 831, Minneapolis, MN, USA) outfitted with hydraulic grips, a 25.4 mm clip on extensometer (2% max strain). Table 1 lists some test parameters, physical properties, and mechanical properties for all the specimens. Tensile dogbone specimens were 150 mm in length. The grip width was 12.7 mm and the gage width was 10.2 mm. The initial peak fatigue stress chosen was approximately 70% of the ultimate strength of specimens tested to failure from the same panel. Note that some of the specimens were subject to increasing peak-stress fatigue steps if they did not fail after at least 24 h. The ratio R was 0.1 and was always maintained for increasing peak stresses. The specimens were also monitored for electrical resistance change, which is not discussed here, having been the major topic of the earlier work [18].

Acoustic emission was monitored using a Digital Wave Fracture Wave Detector (Fracture Wave Detector, Centennial, CO, USA). The test set up is shown in Figure 1. Three wide-band (50–2000 kHz) B1025 Digital Wave AE sensors were applied to the face of the specimen using plastic spring clamps. The outer sensors were separated by 60 mm with the third sensor half way in between. For this AE system, the sensors are all “enslaved”, meaning that, when one sensor triggers, all of the sensors capture and record the waveform on the given channel at exactly the same time. Therefore, for a given event, there are three waveforms. In this way, the data were easily sorted so that only AE data that trigger the middle sensor (sources closest to the middle sensor, i.e., in the gage section of the specimen) were used in the analysis. This was critical since it is not unusual for significant AE activity to emanate from the grips for CMCs. The AE waveform set-up was 1024 or 2048 points, 10 MHz sampling rate, and 25% pretrigger. A 20 dB preamplifier was also used.

Post-test analysis of the AE data was performed using the Digital Wave *WaveExplorer* software (version 7.2) to determine the waveform energies and which of the three sensors triggered event capture, i.e., first arrival. The waveform data were also imported into a MATLAB file to determine the frequency centroid for each waveform and to determine the time of arrival for location analysis using the Akaike information criterion (AIC) method [23,24]. The AIC method analyzes the time-window of the acoustic waveform by determining at each point in time, what precedes and what comes after that given point (in our case the sum of amplitudes) in time, which results in a minimum at the time of the onset of the waveform. The AIC method was found to be more accurate for CMCs than the typical threshold crossing technique [24]. The AE data were sorted according to the events which triggered the center sensor (#3) first, i.e., the gage section, which was used for post-test analysis. The location along the length was determined from the difference in times of arrival between the outermost sensors (#1 and #2) and the speed of sound as determined by events which occurred in the grip region and crossed the outer two sensors during the test [25]. However, when exact location was required for certain sets of data, manual inspection of waveforms to obtain the exact time of arrival from the time of the first peak were employed as described below.

## 3. Results

The stress-step fatigue and AE history are shown in Figure 2. Two of the tests had only one peak stress condition prior to failure, whereas the other three specimens had at least two peak stress steps, as indicated by the stress jumps in Figure 2a. The average energy per event was derived from the average energy on the outer sensors (#1 and #2). The outer sensors are used for energy because the attenuation of high frequency waveforms in these composites has been found to be very high for high frequency events and minimal for low frequency events [26]. In reality, there is little difference between taking the average energy from all three sensors or just the outer two for cumulative AE energy since the cumulative AE energy relationship is dominated by high energy low frequency events. However, only using the average energy for the furthest sensors is more striking for isolating fiber breaks, which are high frequency events. The AE activity is plotted as cumulative AE energy, which has been found to correlate well with transverse matrix crack density [11,12]. Note also that, for the AE activity, there is a jump in energy accumulated for each stress jump, which corresponds to the initial loading of the specimen upon the first cycle of the higher peak stress condition.

Typically, for these types of composites, most of the AE energy occurs on initial loading and dissipates as matrix crack formation decreases or ends with increasing stress. Consequently, the increase in AE energy accumulation is only moderate near failure. The rationale for this is that large transverse matrix cracks which occur over the entire gage section create considerable energy since they are of large area and high modulus. When the crack forms and propagates, considerable surface energy is created from hundreds of cracks (several per mm in multiple plies) during the loading of the specimen. Failure in a typical tensile test for these types of composites would mostly involve fiber failure, which are of significantly smaller cross-sectional area and are only located within a narrow region of the gage section. A striking feature for some of the fatigue specimens in this study is that several of the specimens showed more AE energy cumulated at the end of the test than the entire stress history that precedes it. However, two of the tests did show only a minor increase in AE activity at failure as is typical of these types of composites when tested to failure in tension. It is apparent that the specimens which showed dramatic increase in AE at failure were tested at higher frequencies (two 1 Hz and one 0.1 Hz), whereas the two specimens that did not show this behavior were at lower frequencies (one 0.1 and one 0.01 Hz). To understand what was causing the difference in AE behavior, the methodology for AE analysis described above was employed with some microstructural analysis to correlate AE activity with the damage mechanisms that developed during the tests.

### 3.1. AE Fatigue Cycle Analysis

The set-up for AE acquisition was such that the parametric data (load and strain) were acquired every second in addition to the AE waveforms. Thus, the variation in stress for a given fatigue cycle was not possible for the 1 Hz data. However, for the 0.1 Hz and 0.01 Hz tests, enough load data were available to discern at which point of the cycle the events were occurring. The AE analysis for the 0.1 Hz test (113-3) are presented here in detail, which showed the enhanced AE activity near failure (Figure 2b). The features found for the 0.1 Hz test were also observed for the 1 Hz tests (147-3 and 113-9) for the most part. Although both specimens were tested at 1 Hz, the same acquisition rate for load in the software, the peak and valley events could be discerned by the timing of the event. In other words, the peak and valley events occurred approximately 0.5 s from one another. Where there were differences is discussed below.

The AE data were separated into events which occurred near the peak of the cycle and events which occurred near the valley of the cycle. Figure 3a shows the cumulative AE energy data for the overall test as well as the contributions of energy for the peak and valley. There were no valley events until the 201,990 s (56.1 h) mark of the test, 2.4 h prior to failure. After the occurrence of valley events, most of the AE energy was dominated by what occurred during the valley part of the stress-cycles with some appreciable AE occurring during the peak of the cycle as well. An example of when the valley and peak events occur in a stress cycle is shown in Figure 3b for the cycle centered at 210,001 s. Note that “valley” events occur during unloading as the valley is approached and “peak” events occur upon loading as the peak of the stress-cycle is approached.

To understand the intensity and progression of peak and valley events, an energy analysis was performed. For each event the average energy was determined by taking the average of the energies of the two outer sensors. Figure 4 shows the AE event energy versus time for the entire test that were added in sequence to produce Figure 3. Each data point represents a single AE event from a discrete source. Significant AE activity occurs at the beginning of the test, primarily in the first few cycles. For most of the test, little AE occurs until the very end of the test. The energy ranges from 0.1 to 50 V^2^/μs until near the end of the test where energies less than 0.01 V^2^/μs were recorded. Since the valley events were only observed at the end of the test (201,990 s), the peak and valley events were plotted separately after 200,000 s in Figure 4b,c, respectively. There were 2156 valley events compared to 1367 events over this period. Valley events were also much more prevalent prior to about 206,000 s. It is also obvious that the valley events tended to be of higher energy than the peak events though both peak and valley showed increasing energy events as time progressed. The peak events contained low and high energies (Figure 4b), including nearly all the very low energy events at the end of the test.

To understand the relationship of valley and peak AE events, the number of events per cycle were counted for valley and peak events for the time range greater than 200,000 s (Figure 5). It should be noted that, prior to this time, most cycles did not even have one AE event except for the beginning of the test. Figure 5 shows that there were more valley events per cycle than peak events until very near failure (~12th last cycle). Therefore, it is apparent that whatever is happening upon unloading causes or at least contributes to what happened upon loading during the fatigue cycles. Since there is a preponderance of unloading (valley) events at the onset of this high AE activity region approaching the end of the test (Figure 4c), it appears to precipitate the subsequent intensity of sources causing AE during the loading (peak) part of the cycle. To better understand the nature of what is causing the AE events, a location and frequency analysis was performed.

### 3.2. AE Event Location Analysis

Location was determined from the difference in times of arrival of the two outer sensors, Δt. In addition, the speed of sound of the extensional wave or the time it takes for sound to travel between the outer sensors, Δt_x_, was required, where x is the distance between the two outer sensors. The speed of sound (Δt_x_/x) increases throughout the test as damage develops [6,24]. This was accounted for by determining the Δt_x_ from events which occurred outside the outer sensors, presumably in the grips [25]. There was a period of no AE activity between 100,000 and 130,000 s (Figure 4a). For the following, only the location data after 130,000 s is presented since it leads up to failure. The location of the events prior to 100,000 s were evenly distributed across the gage section. The AIC derived Δt values typically differed less than 1 μs of the actual first peak Δt. However, to get the most accurate Δt, the times of arrival of the first peaks on sensors 1 and 2 for events which occurred after 130,000 s were determined manually [13] and used to determine location. Examples of valley and peak waveforms are shown in Figure 6 with arrows indicating first peaks for sensors 1 and 2, top and bottom, respectively. Note that the first peak for the valley event on the top and bottom sensor is negative and the first peak for the peak event is positive; this is significant and is explained below.

From the determination of first peak times of arrival, the location was determined from:
Location = (x/2)(Δt/Δt_x_)(1)
where x is the length between the outer sensors (25 mm). The location of each event after 130,000 s is plotted in Figure 7a versus the time of the test where the value 0 is the center of the gage section. Each data point represents a single event and the width of the data point is proportioned to the average AE energy of the given event. There are scattered and infrequent peak events for the over 70,000 s (~20 h) leading up to the heightened period of AE activity just prior to failure. The events at the end of the test are so dense that the period after 200,000 s (box in Figure 7a) are plotted in Figure 7b,c for the valley and peak events, respectively. Most of the peak and all of the valley events are concentrated in the 0 to +1.5 mm location from the center of the gage, which is where failure took place. Whatever occurred just prior to failure would not correspond to distributed matrix cracking along the gage length. Therefore, one would expect to observe localized damage near the fracture surface within an approximate 2 mm length or less of the composite, as shown below.

### 3.3. AE Waveform Analysis

In earlier studies [16,17,18,27], it was found that frequency content was often helpful in identifying some microfracture mechanisms such as matrix cracks and fiber fractures as well as the location of matrix cracks, e.g., surface versus internal. The frequency centroid has been a useful parameter to assess the frequency content of waveforms for this purpose [18,20]. Figure 8 shows the waveform and Fast Fourier Transform (FFT) for the same events as Figure 6 except for the middle sensor 3. It has been shown that higher frequencies do not transmit over longer distances effectively in these types of composites [26], so the sensor closest to the source (middle) was used for frequency analysis. The waveform/FFT of the two events for the frequency centroid (FC) is also shown in Figure 9. Note that the valley event (Figure 8a) has a frequency centroid about 200 kHz lower than that of the selected peak event (Figure 8b). The frequency centroids of all the valley and peak events are plotted in Figure 9 for the last few hours of the test.

Valley events dominate at first near the end of the test (~202,000 s) and increase in number, frequency range (especially lower frequency content) and energy range (especially higher energy events) with time. Valley events are also characterized by the first peak being negative (Figure 6a) when measured in the far field (sensors 1 and 2). This points to whatever is happening in the valley as being compressive. When a compressive force causes a microfracture event such as a longitudinal crack, something being crushed, or a shear crack, the front of the waveform, an extensional wave, will be in tension (negative direction for surface transducer). In large semi-infinite structures such as bridges or the earth, AE events emanating from shear can give positive or negative first peaks depending on the location of the sensor with respect to the shear direction (if ahead of the source and in the shear direction, positive, and if behind the source and opposite the shear direction, negative) [28]. However, for a plate, the extensional wave is not like a P-wave in the semi-infinite case and the direction of the first peak would be dictated by the direction of strain response. For example, if a transverse crack forms in tension or a fiber breaks in tension, the direction of material response (spring back) would be outward, away from the original location, and “push” the material in compression outward resulting in the front end of the stress wave being positive. The opposite would be true for a compressive mechanism regardless of the mechanism itself (e.g., longitudinal split, shear or crushing). All of the valley events showed this negative first peak on the far sensors. Note that the near sensor in Figure 8a shows a positive first peak for the valley event. Some valley events showed positive and other events showed negative first peak. The middle sensor is essentially on top of the source. Consequently, the first peak is dominated by the flexural wave since the extensional part of the waveform cannot separate itself from the flexural part of the waveform over the short distance from the source.

Therefore, the sign of the first peak on the middle sensor probably is indicative of the direction of flexure of the specimen caused by the stress wave and is perhaps related to the “side” of the specimen that the damage event occurred.

Peak events at the end of the test are characterized by a positive first peak (Figure 6b). This would be indicative of a tensile force causing the microfracture event resulting in the front of the extensional component of the stress wave being in compression. For this specimen, peak events occur later in time (>205,000 s) than the onset of the valley events. This means that a significant number of valley events occurred for about an hour prior to the onset of significant peak event formation. The number, frequency range and energy range of the peak events also increase with time as failure is approached (Figure 9b). The peak event frequency range, however, is much broader and there are several very high frequency content events prior to failure.

### 3.4. Microstructural Analysis

A polished longitudinal section 113-3 is shown in Figure 10 for a section far from the fracture surface and at the fracture surface. Near the fracture surface, there are considerable cracking phenomena. Not only are there transverse cracks, but there are also a significant number of longitudinal and diagonal oriented cracks, especially in the 90 fiber tow regions. Some arrows are provided to highlight these types of cracks. This more intense region of damage is on the order of 1 mm from the fracture surface itself. For the polished section far from the fracture surface, the average transverse crack spacing is ~1 crack per mm and there is no evidence for concentrated damage in 90 fiber tow regions. It is also interesting that there is significant transverse matrix cracking near the fracture surface with much smaller crack spacing than the far field. Therefore, it is reasonable to assume that the AE events which occur near the fracture surface location correspond to the increased damage observed in the polished section.

Two inferences can be made from these observations:The unique damage to the fracture surface region appears to be associated with 90 tow longitudinal and diagonal shear type cracking. The 90 tow region is the weakest region within these types of composites since the boron nitride (BN) interphase that separates the fibers from the chemical vapor infiltrated (CVI) portion of the matrix is a weak layer resulting in easy crack propagation parallel to the fiber direction. In addition, there is porosity within the tow which would offer stress-concentrator sites for crack initiation under a variety of local stress states. This would be in-line with valley AE events observed due to compressive forces since a compressive force would most likely initiate the longitudinal and diagonal shear type cracking rather than tensile.The increased density of transverse cracks propagating especially through the 0 tow regions and melt-infiltrated (MI) matrix regions near the fracture surface appear to have strong interaction with the 90 tow cracking. Since the 90 tow regions are weaker, it would imply that they occur first and are the initiators of the transverse cracking–the same progression as would be inferred from valley AE occurring first and stimulating peak AE (local transverse cracking and/or fiber breakage) in Figure 5 and Figure 7.

### 3.5. Comparison to 1 Hz and 0.01 Hz Fatigue

The two 1 Hz fatigue tests were similar to the 0.1 Hz fatigue test described above. The 1 Hz AE events had the same first peak directional character for peak (positive) and valley (negative) events. There are far more valley AE events and cumulative energy at the end of the test compared to peak AE activity. All events prior to the end of the test when valley events were observed were peak events. The frequency content and energy range behavior for valley events at the end of the test followed the same trend as for 113-3.

Figure 11 shows the cumulative AE energy and location of peak and valley events for 147-3. There is one apparent difference between 147-3 (1 Hz) and 113-3 (0.1 Hz). For the 147-3, a significant number of peak events (Figure 11c) occur prior to the onset of the valley events starting at ~84,600 s (Figure 11a). Most of these events are low in energy and high in frequency. The valley events were centered at +6.25 mm location (Figure 11b), whereas the peak events were centered at ~7.5 mm (Figure 11c). This corresponds with the region of fracture where there were two main “planes” for the fracture surface approximately 1 mm apart (Figure 12a). Some regions of longitudinal and diagonal cracking in the 90 tows as in the 0.1 Hz specimen were observed near the fracture surface on the lower half of the fracture surface shown in Figure 12a, which is a polished section close to the original edge of the tensile bar. The specimen was cut so that the surface longitudinal plane approximately 3.5 mm from the edge (Figure 12b) could be observed as well. Longitudinal cracking was pervasive near the fracture surface across the width in this region. Transverse matrix cracking through the 0 tow regions and MI matrix, which appears to emanate from 90 tow cracks, was prevalent in both regions.

Since this specimen just prior to failure first had peak events occur in the failure region, most with high frequency and low energy, it is likely that these events corresponded to fiber breaks, as in other studies [16,27]. This would infer that local fiber breakage resulted in altering the stress-state in the specimen (for example if fiber breakage occurred on one half of the specimen) in the soon-to-be failure location which triggered the onset of “compressive” mechanisms which would then exacerbate peak stress damage leading to ultimate failure. This may also be occurring in 113-3 as well over a longer time due to the lower frequency; however, it appears more pronounced in 147-3.

For 0.01 Hz fatigue, valley events were not observed prior to failure. Figure 13a shows the AE activity of the last stress step leading to failure with significant AE activity in the failure region of +2.5 mm. Most of the events in this region are also low in energy (smaller bubble size in Figure 13a) and high in frequency (Figure 13b). This would infer a more common failure process where fiber breakage would cumulate locally leading to ultimate failure. Note that the fracture surface (Figure 14) does not show significant 90 tow damage other than one 90 longitudinal crack compared to the other two specimens (Figure 10 and Figure 12).

## 4. Discussion

This study demonstrated different damage development, illuminated by the aid of acoustic emission, leading to failure for different fatigue frequency conditions on similar composites. The quality of the composites themselves were not very good including significant porosity; however, the methodology and findings are significant enough to merit further study and application to more advanced composite systems.

Essentially, three different damage developments were observed. For the two composites examined subject to higher frequencies (0.1 and 1 Hz), damage development included microfracture events that occurred in the unloading part of the fatigue cycle near or at the stress minimum (valley) which exacerbated damage development on subsequent loading and peak portions of the fatigue test just prior to failure. The valley events only occurred in the region that was to fail within a few hours of ultimate failure. These valley events produced more AE energy at the very end of the test than all the AE events prior—a factor that appears to be a good indication that a valley mechanism is at work. However, what triggered the onset of valley events appears different for the two different experiments. The 0.1 Hz specimen appeared to have valley events form with little prior AE activity in the soon-to-be failure region so it was not clear what initiated the onset of valley events. The 1 Hz specimen had a significant number of high frequency peak AE events, presumed to be fiber breakage, leading up to the onset of valley events. The third type of fatigue failure did not have any valley events just prior to ultimate failure (0.01 Hz specimen) with significant low energy high frequency AE occurring near the peak of the stress cycle presumed to be cumulative fiber breakage leading to ultimate failure.

An attribute of the valley event waveforms that were captured by sensors a significant distance from the source was that the direction of the first peak was negative. This would mean that the front end of the extensional wave was dilated (tensile). This could only be caused by an event that was in compression. Microscopy showed that, at and near the fracture surface, longitudinal and diagonal cracks were formed within the 90 tow regions (made up of the SiC fibers, BN interphase and CVI SiC sheath) of the woven architecture. These would correspond to local interlaminar and shear microfracture events within the 90 tow regions, respectively. The result of these local damage in the 90 tow regions would then result in transverse cracking in the 0 tow region and MI matrix due to the presence of new local transverse crack initiation sites resulting from the 90 tow damage as well as the redistribution of stress within the composite as the 90 tow region would effectively lose load-carrying ability.

A related issue to the valley mechanisms is its compressive nature while the composite was actually still subject to a global tensile stress (R = 0.1). For SiC/SiC MI composites, it is typical for the matrix to be in compression due to the increase in volume that occurs when the liquid phase Si changes to solid Si [7]. After transverse matrix cracking at some tensile stress resulting in non-linear stress behavior and a hysteresis unload/reload stress strain behavior, materials with residual compression in the matrix exhibit stiffening in the unloading curve at low stresses [29]. The cause of the stiffening is due to crack closure upon unloading because, when the matrix relieves the compressive stress at a matrix crack, the matrix is free to expand resulting in the two halves of the matrix contacting one another as stress approaches zero. Therefore, for these types of composites, it is not surprising that a compressive force could be exerted in the region of the matrix crack on the matrix upon unloading. In addition, this would be enhanced by fatigue where the interfacial shear stress at the interface within the fiber/matrix interphase debonded region would decrease with fatigue cycles due to wear. This would result in increased volume of matrix that could relieve its compressive residual stress [30]. In addition, the redistribution of local stress due to local fiber breakage and debris from interfacial wear could produce local high stress contact points during unloading and contribute to local 90 tow microfractures in compression. Of course, the degree of compressive force upon unloading would depend on the R ratio and perhaps composite system. One would expect higher R ratios to not have the compressive types of mechanisms and negative R ratios to have worsening of the compressive mechanisms. Similarly, the reason the 0.01 Hz specimen (111-9) did not have any valley events may have been due to the higher peak fatigue stress at which it failed. The higher minimum global tensile stress value for higher peak stress due to R = 0.1 would minimize or negate local compressive sites as the minimum stress is approached. Porosity was also significant in these composites and may also have played a role in the relief of matrix stress and/or creation of debris which may have fostered the valley mechanisms. More study is required to determine if the valley event mechanism occurs or not in denser state-of-the-art woven MI composites.

## 5. Conclusions

Acoustic emission was shown to be an excellent technique to identify the progression of damage accumulation in fatigue tests performed on woven SiC fiber reinforced SiC based matrix composites under three different frequency conditions. Failure in some cases was influenced by damage that occurred during the unloading portion of the fatigue cycle as well as fiber breakage during the loading to peak portion of the fatigue cycle. AE was able to discern the different progression and types of damage via location, energy, frequency and direction of the first extensional peak of the waveform. The use of the latter AE waveform feature, direction of first extensional peak, was novel for these kinds of tests and may be another useful feature to discern damage mechanisms when using AE for materials and structures testing.

## Figures and Tables

**Figure 1 materials-11-02477-f001:**
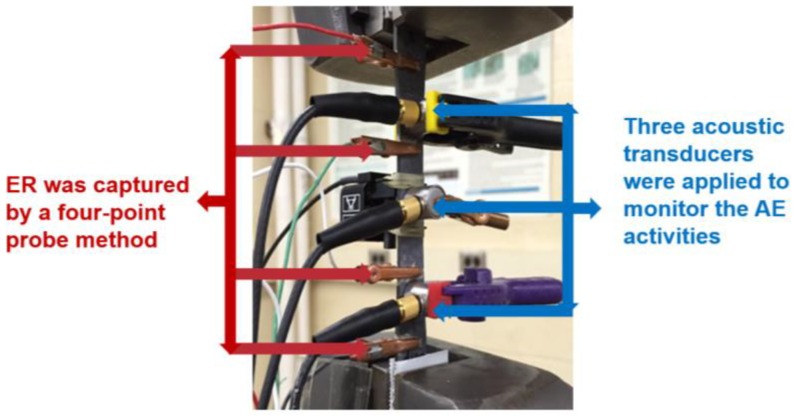
Test set-up for tensile fatigue tests showing three AE sensors and electrical resistance leads (not part of this study).

**Figure 2 materials-11-02477-f002:**
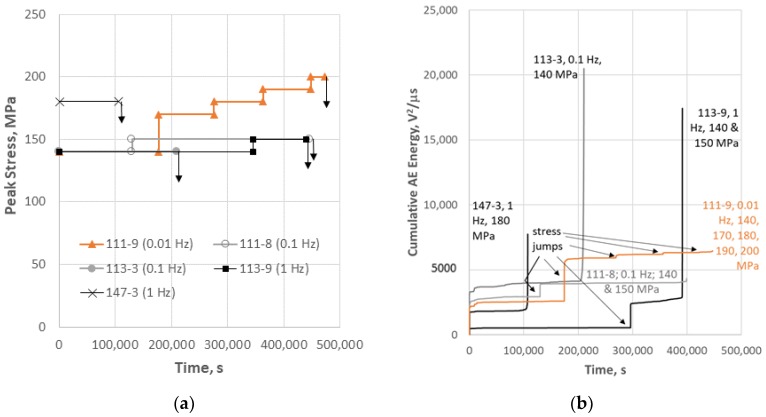
The history of stress condition and AE activity for the CMC specimens: (**a**) peak stress plotted versus time for R = 0.1 fatigue cycles where the downward arrow indicates failure; and (**b**) cumulative AE energy versus time for each specimen.

**Figure 3 materials-11-02477-f003:**
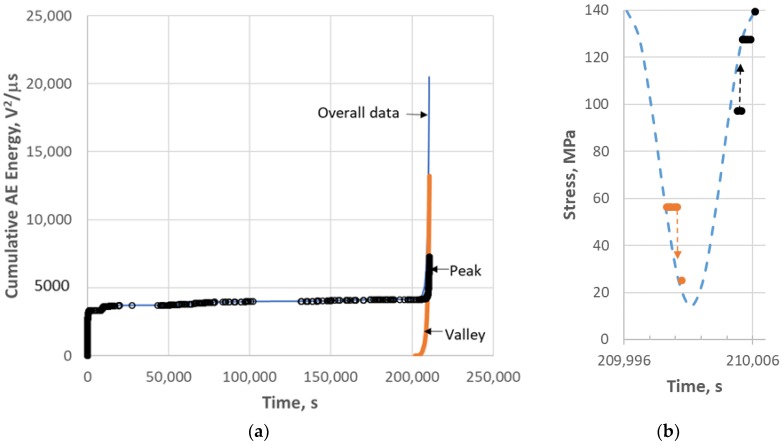
(**a**) Cumulative AE energy for 113-3 (0.1 Hz fatigue). The “Overall data” are separated into the part of the data that occurred during the peak of the fatigue cycles and the data that occurred during the valley. (**b**) A stress cycle near the end of the test showing the occurrence of the “valley” (unload) events and the “peak” (reload) events. Note that the AE acquisition software only captured the load data every second. Thus, for 1 s the load acquired for a given event within that 1 s period was the same as other events acquired within that period, even though the load was either increasing or decreasing during that period. The actual stress of a given event to the right of the stress curve should be displaced downwards for valley events and upward for peak events onto the stress line.

**Figure 4 materials-11-02477-f004:**
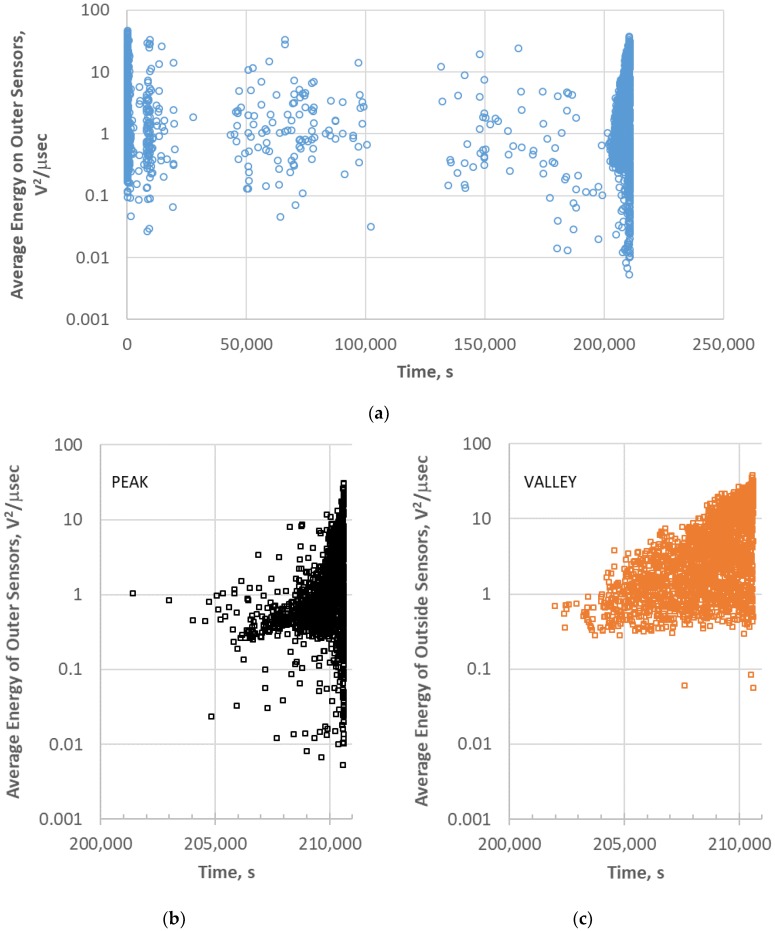
Average energy per event of 113-3: (**a**) for the entire test; (**b**) for peak events at the end of the test; and (**c**) for valley events at the end of the test.

**Figure 5 materials-11-02477-f005:**
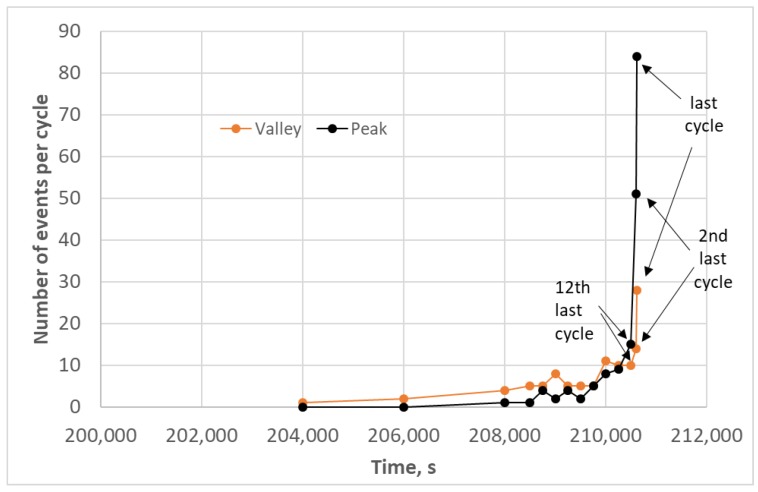
Number of events per cycle for peak and valley events for 113-3 at the end of the test.

**Figure 6 materials-11-02477-f006:**
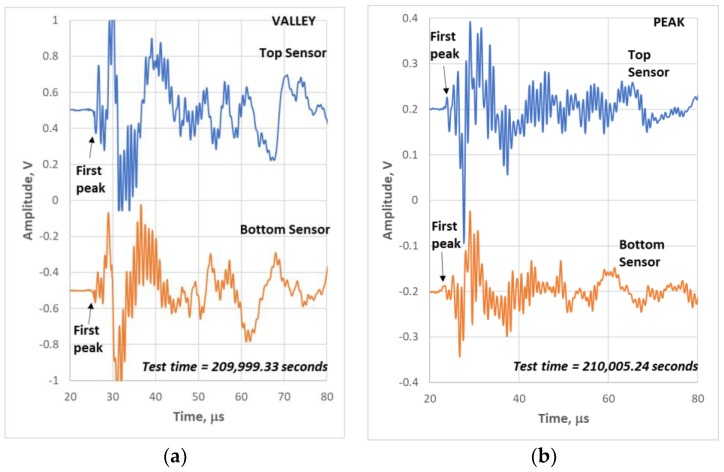
Waveforms from top and bottom sensors for: (**a**) a valley event; and (**b**) a peak event showing the first peaks of each waveform from 113-3.

**Figure 7 materials-11-02477-f007:**
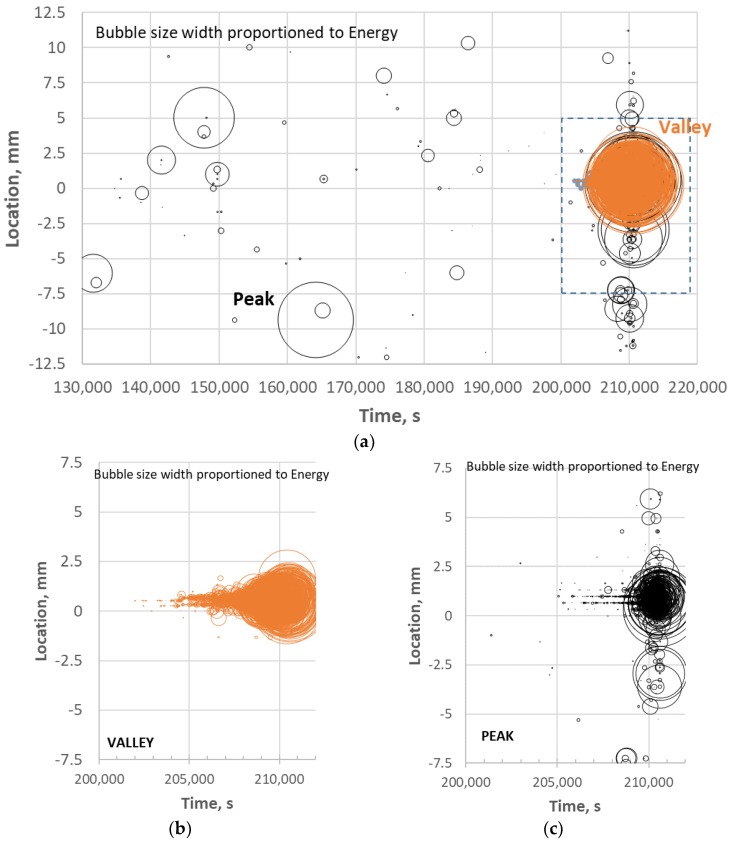
AE location analysis distinguishing between “peak” and “valley” events for: (**a**) the last half of the test; and for the (**b**) valley and (**c**) peak events for the last ~3 h of the test for 113-3.

**Figure 8 materials-11-02477-f008:**
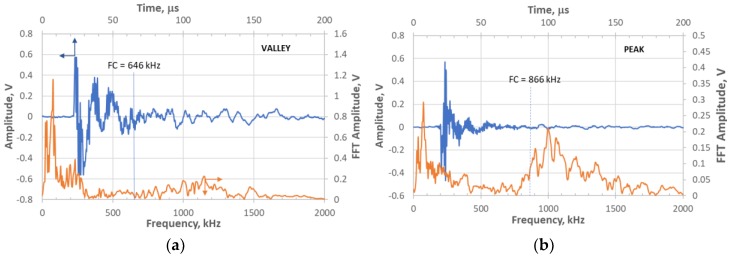
Waveform and Fast Fourier Transform (FFT) of sensor 3 waveform for the same waveforms in Figure 6: (**a**) valley event at 209,999.33 s; and (**b**) peak event at 210,005.24 s during the test.

**Figure 9 materials-11-02477-f009:**
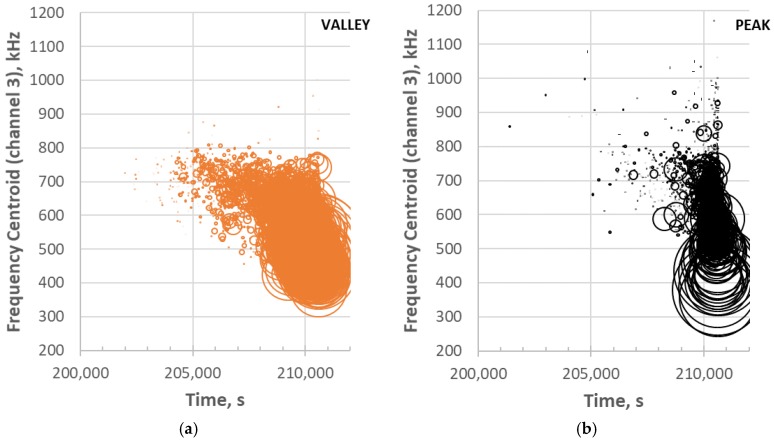
Frequency centroids for (**a**) valley and (**b**) peak events at the end of the test. Note that the bubble width is normalized to AE energy of the waveform.

**Figure 10 materials-11-02477-f010:**
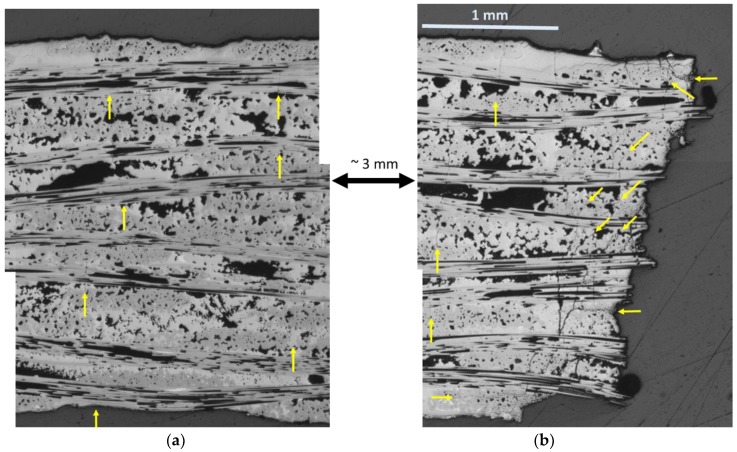
Polished longitudinal sections of 113-3 CMC specimen (**a**) far from fracture surface and (**b**) near fracture surface showing increased damage regions near fracture surface.

**Figure 11 materials-11-02477-f011:**
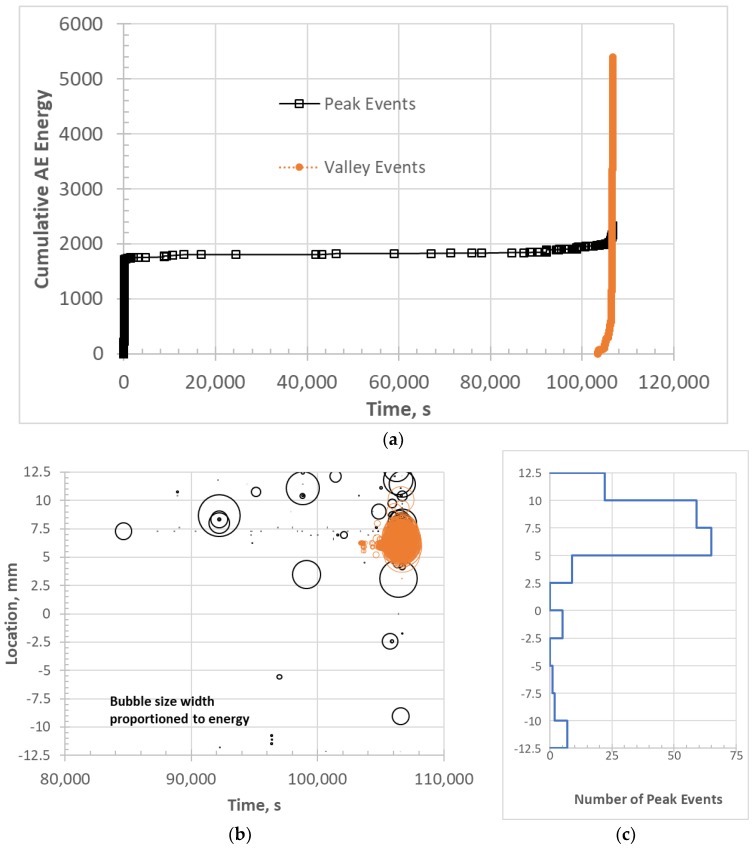
AE activity for 1 Hz fatigue specimen 147-3: (**a**) cumulative AE energy contributions from peak and valley events; (**b**) location of peak and valley events during the last part of the test; and (**c**) histogram of peak event locations along the gage length.

**Figure 12 materials-11-02477-f012:**
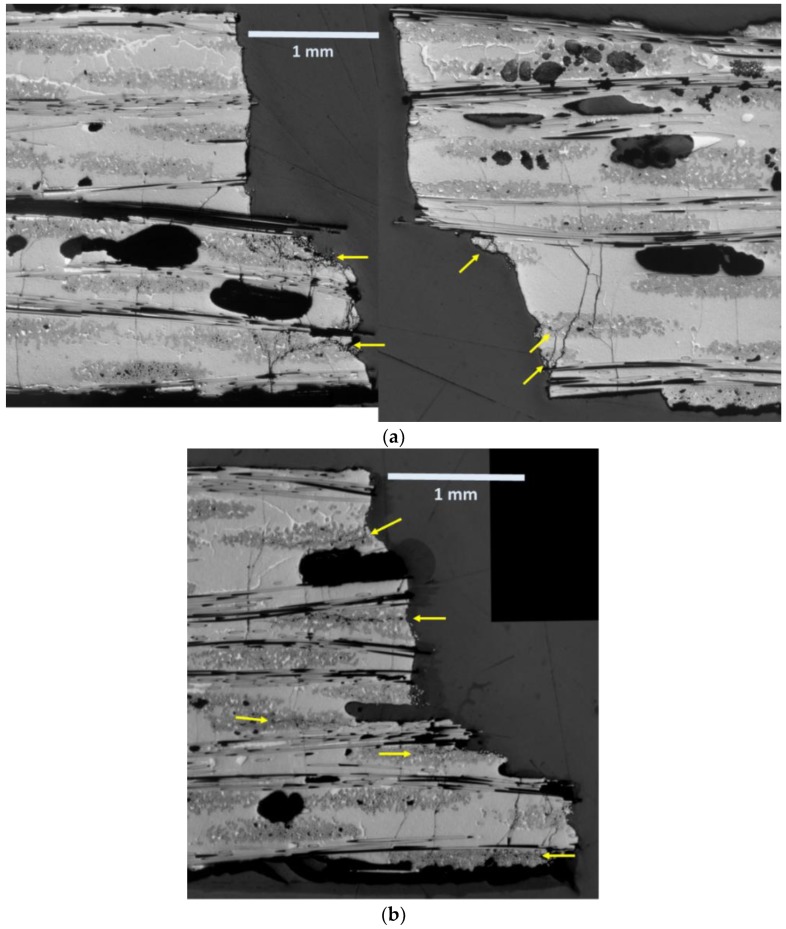
Polished longitudinal sections of 147-3: (**a**) both sides of the fracture surface approximately 1 mm from the edge; and (**b**) the left side of the fracture surface approximately 3.5 mm from the edge of the specimen. Arrows indicate some of the longitudinal and diagonal cracking in the 90 tows.

**Figure 13 materials-11-02477-f013:**
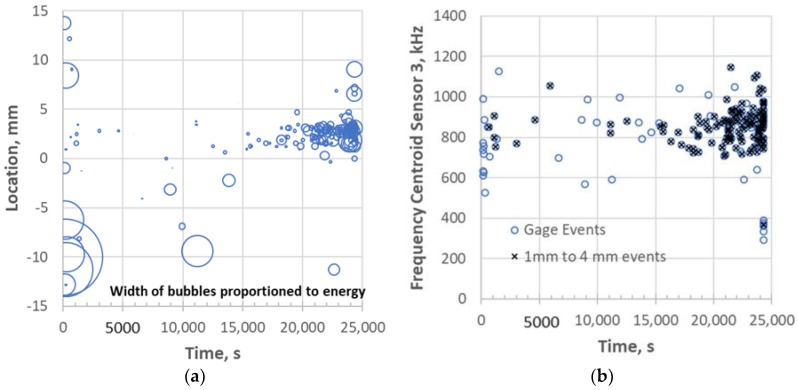
Acoustic emission for 0.01 Hz tested 111-9 specimen: (**a**) location of AE at end of test; and (**b**) frequency centroid of gage events at end of test.

**Figure 14 materials-11-02477-f014:**
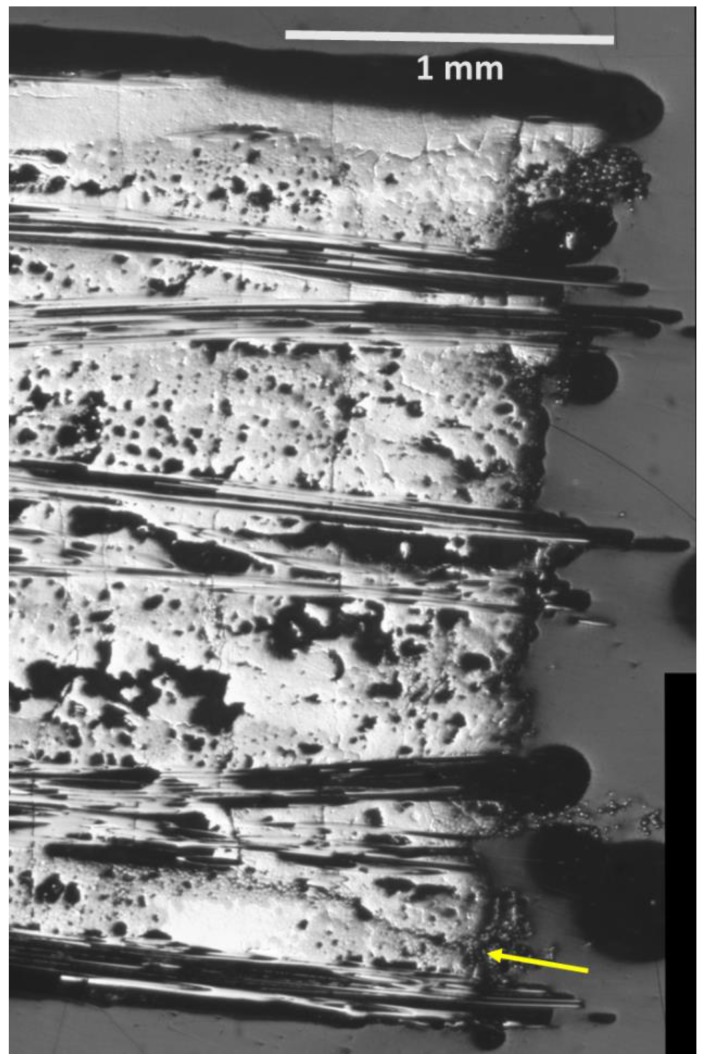
Fracture surface of 111-9.

**Table 1 materials-11-02477-t001:** Physical and Mechanical Properties of Specimens Tested.

Specimen	Thickness, mm	Fiber Volume Fraction	E, GPa	Frequency, Hz	Peak Fatigue Stress per Step, MPa
111-8	3.0	0.238	206	0.1	140; 150
111-9	2.9	0.26	221	0.01	140; 170; 180; 190; 200
113-3	3.0	0.236	185	0.1	140
113-9	2.8	0.246	226	1	140; 150
147-3	2.9	0.258	248	1	180

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
