# Peer review of "Damage Determination in Ceramic Composites Subject to Tensile Fatigue Using Acoustic Emission"

_materials, 2018, doi:10.3390/ma11122477_

Round 1

Reviewer 1 Report

Please check the following lines - there are meaning or format errors:

56 ...initial microcracks are typically very small (e.g., 90 two tunnel cracks [8]),

59 ...creep-rupture [9], i.e., the flaw that would initiate a slow-crack growth mechanism controlled failure 60 crack at elevated temperatures under a constant stress load.

In Figure 2, it'd help to use identical time axes in both graphs

303 ...travel to separate itself from the flexural (the wavelengths of the extensional...

439 A related issue to the valley mechanisms is there compressive nature

467 Funding: Please add: This research was partially funded

Author Response

56 ...initial microcracks are typically very small (e.g., 90 two tunnel cracks [8]), - DELETED “TWO”

59 ...creep-rupture [9], i.e., the flaw that would initiate a slow-crack growth mechanism controlled failure 60 crack at elevated temperatures under a constant stress load. – DELETED LAST PART OF SENTENCE. THE SIGNIFICANCE IS THAT THE STRESS FOR INITIAL MICROCRACKING, WHICH CAN ONLY BE DETECTED BY AE, CORRELATES WITH LONG TIME (~ 1000 HR) STRESS-RUPTURE FAILURE STRESS.

In Figure 2, it'd help to use identical time axes in both graphs - CORRECTED

303 ...travel to separate itself from the flexural (the wavelengths of the extensional...RE-WROTE, NOW LINES 335 TO 338

439 A related issue to the valley mechanisms is there compressive nature – CHANGED “THERE” TO “ITS”

467 Funding: Please add: This research was partially funded - CORRECTED

Reviewer 2 Report

The authors present here an interesting study on AE observations of SiC/SiC composites. First of all, the text should be checked better in order to perform some English corrections and in order to make the text more clear in many cases. Moreover, the following points should be addressed prior to publication.

Abstract: Seems like a generic description of AE, the objective and the contribution of this work have to be more clearly stated, it needs adaptations.

Line 29: Two times the word ‘such’, remove one time.

Lines 30-38: Can the authors explain why, according to them, this is a special mechanism occurring only in CMCs?

Lines 45-47: Either the present or the past tense should be used in all sentences.

Line 56: 90 two tunnel cracks: what do the authors mean?

Line 67: In for: erase one of the two.

Line 72: In a SiC/SiC composite laminate.

Generally, the authors give a satisfactory list of references related to AE studies in CMCs. However, the authors don’t describe clearly the problem statement and what is the target of their contribution towards possible problems in the introduction. The objectives have to be clearly stated.

Line 104: Please specify amount of time.

Table 1: 1) Fiber fraction refers to volume or mass? 2) Peak stress: is this the maximum fatigue stress? The ultimate stress of the corresponding laminates should be given so that the percentage of the maximum fatigue stress is known (as % of the ultimate static strength). The reasoning of the choice should be explained. 3) The title should be modified: testing parameters are also presented here, not only properties.

Line 126: Can the authors give some more information on how location of the events was performed? Was attenuation taken into account? What is the amplitude threshold applied?

No information regarding the geometry of the tested specimens is given in the corresponding section. This information should be added.

Figure 2: Please use either seconds or hours in both graphs. Sometimes the authors use Hz and sometimes hz. Also, what do the units of the AE energy represent?

Line 170: It is not clear what the authors want to say in the note.

Lines 187-190: It is not clear what the authors want to say here.

Lines 193-194: Why only the two outer sensors are taken into account?

Figures 6 and 7 related to the location analyses are clear, however the location methodology is somehow confusing. It is not clear how specific events from the grips were located, how the wave velocity was taken into account and what Δtx corresponds to. 

Line 285: Like Figure 6 and not 7.

Figure 10: Can the authors add a figure of the cross section further away from the fracture area in order to prove the said differences? Also can the authors add a picture of the failed specimen?

Line 346: Can the authors explain then what the initial peak signals throughout the fatigue test and before the valley signals correspond to?

Line 364: The difference between specimens with different frequencies doesn’t seem to be reasonable since in both cases there are peak events recorded prior to valley events from the beginning of the test. So it is not clear what the authors mean here.

Line 439: Their instead of there?

Line 441: Due to.

As a general comment regarding the discussion, the authors haven’t performed any correlation with the applied stress for each composite. Don’t the authors believe that the stress level is also associated with differences in AE?

Author Response

The authors present here an interesting study on AE observations of SiC/SiC composites. First of all, the text should be checked better in order to perform some English corrections and in order to make the text more clear in many cases. Moreover, the following points should be addressed prior to publication.

Abstract: Seems like a generic description of AE, the objective and the contribution of this work have to be more clearly stated, it needs adaptations. – ADDED KEY FINDINGS OF DAMAGE PROGRESSION FOR FATIGUE AT DIFFERENT FREQUENCIES TO THE END OF ABSTRACT

Line 29: Two times the word ‘such’, remove one time. – REMOVED A “SUCH”

Lines 30-38: Can the authors explain why, according to them, this is a special mechanism occurring only in CMCs? – I DID NOT STATE THAT IT IS ONLY OCCURRING IN CMCS BUT THAT IT IS THE MECHANISM FOR NON LINEAR STRESS STRAIN IN CMCS UNDER TENSION. PERHAPS I AM MISSING THE POINT OF THE REVIEWER?

Lines 45-47: Either the present or the past tense should be used in all sentences. CORRECTED A NUMBER OF INSTANCES

Line 56: 90 two tunnel cracks: what do the authors mean? – DELETED “TWO”

Line 67: In for: erase one of the two. – DELETED “IN”

Line 72: In a SiC/SiC composite laminate. – I CHANGED IT TO “SIC/SIC LAMINATE COMPOSITE”. THIS IS TO DISTINGUISH WITH WOVEN COMPOSITES.

Generally, the authors give a satisfactory list of references related to AE studies in CMCs. However, the authors don’t describe clearly the problem statement and what is the target of their contribution towards possible problems in the introduction. The objectives have to be clearly stated. – ADDED TO LAST PARAGRAPH OF INTRODUCTION (LINES 94-99) ON INFORMATIVE NATURE OF AE TO DAMAGE PROGRESSION IN THESE COMPOSITES

Line 104: Please specify amount of time. – “AT LEAST TWENTY FOUR HOURS ADDED IN TEXT”

Table 1: 1) Fiber fraction refers to volume or mass? 2) Peak stress: is this the maximum fatigue stress? The ultimate stress of the corresponding laminates should be given so that the percentage of the maximum fatigue stress is known (as % of the ultimate static strength). The reasoning of the choice should be explained. 3) The title should be modified: testing parameters are also presented here, not only properties. – TABLE AMENDED FOR “VOLUME” FRACTION, CLARIFIED “PEAK FATIGUE STRESS PER STEP”, INCLUDED SPECIMEN THICKNESS AND INCLUDED SENTENCE ON FATIGUE STRESS CHOSEN BASED ON 70% OF UTS (LINES 116-117)

Line 126: Can the authors give some more information on how location of the events was performed? Was attenuation taken into account? What is the amplitude threshold applied? – EXPANDED EXPLANATION AS TO USE OF AIC METHOD AND NOT THRESHOLD CROSSING (LINES 141-154)

No information regarding the geometry of the tested specimens is given in the corresponding section. This information should be added. – ADDED TO IN LINE 116 – 150MM LONG, 10.2 MM WIDTH IN GAGE SECTION

Figure 2: Please use either seconds or hours in both graphs. Sometimes the authors use Hz and sometimes hz. Also, what do the units of the AE energy represent? - CORRECTED

Line 170: It is not clear what the authors want to say in the note. – I ADDED THAT THE ACQUISITION RATE FOR THE LOAD DATA AND THE STRESS CYCLE ARE THE SAME – 1 HZ – THE POINT IS THAT THE PEAK AND VALLEY STRESS WOULD BE 0.5 SECONDS APART WHICH COULD BE DISCERNED FROM THE EVENT TIME

Lines 187-190: It is not clear what the authors want to say here. – 217 to 220 ADDED SENTENCES EXPLAINING ACQUISITION OF LOAD IN AE SOFTWARE AND ITS RELATION TO ACTUAL STRESS

Lines 193-194: Why only the two outer sensors are taken into account? – THIS WAS ADDED AT BEGINNING OF SECTION 3, LINES158 TO 164 – MAINLY THIS WAS DONE BECAUSE HIGH FREQUENCY EVENTS ATTENUATE CONSIDERABLY COMPARED TO LOWER FREQUENCY (HIGH ENERGY) EVENTS. IT IS USEFUL FOR IDENTIFYING FIBER BREAKS USING LOW ENERGY OF FAR SENSORS AND FREQUENCY FROM MIDDLE SENSOR. AVERAGE ENERGY FOR HIGH ENERGY EVENTS, WHICH DOMINATE THE CUMULATIVE ENERGY CURVE, IS NOT THAT AFFECTED BY USING OUTER SENSORS OR ALL THREE SENSORS.

Figures 6 and 7 related to the location analyses are clear, however the location methodology is somehow confusing. It is not clear how specific events from the grips were located, how the wave velocity was taken into account and what Δtxcorresponds to. – REFERENCES 6, 24 AND 25 WERE GIVEN THAT HAVE SHOWN THIS IN THE PAST. ESSENTIALLY WELL FORMED EXTENTIONAL WAVES AND MAX DTx INDICATE EVENTS OUTSIDE THE SENSORS WHICH ARE PROBABLY IN OR NEAR THE GRIPS

Line 285: Like Figure 6 and not 7. - CORRECTED

Figure 10: Can the authors add a figure of the cross section further away from the fracture area in order to prove the said differences? Also can the authors add a picture of the failed specimen? – A FAR FIELD IMAGE WAS ADDED SHOWING 1MM CRACK SPACING, TYPICAL OF ENTIRE FAR FIELD. THE SPECIMENS WERE POLISHED AND NO FRACTURE SURFACE IS AVAILABLE

Line 346: Can the authors explain then what the initial peak signals throughout the fatigue test and before the valley signals correspond to? –IN LINES 383 AND 384 ADDED “TRANSVERSE CRACKING AND/OR FIBER BREAKAGE” REFERING TO PEAK SIGNALS WHICH FOLLOW VALLEY SIGNALS

Line 364: The difference between specimens with different frequencies doesn’t seem to be reasonable since in both cases there are peak events recorded prior to valley events from the beginning of the test. So it is not clear what the authors mean here. – GOOD POINT, ESPECIALLY WHEN CYCLES ARE CONSIDERED RATHER THAN TIME. I ALL IT “APPARENT” DIFFERENCE IN LINE402 AND LATER IN LINES 420-421 SAY IT MAY BE OCCURING IN 113-3 AS WELL, JUST NOT AS NOTICABLE SINCE IT IS AT A LOWER FREQUENCY.

Line 439: Their instead of there? - CORRECTED

Line 441: Due to. - CORRECTED

As a general comment regarding the discussion, the authors haven’t performed any correlation with the applied stress for each composite. Don’t the authors believe that the stress level is also associated with differences in AE? – ANOTHER GOOD POINT AND COULD EXPLAIN WHY NO VALLEY EVENTS FOR THE 0.01 HZ SPECIMEN. I ADDED TWO SENTENCES IN THE DISCUSSION (LINES 497-500): “Similarly, the reason the 0.01 Hz specimen (111-9) did not have any valley events may have been due to the higher peak fatigue stress at which it failed. The higher minimum global tensile stress value for higher peak stress due to R = 0.1 would minimize or negate local compressive sites as the minimum stress is approached.”

Reviewer 3 Report

The paper deals with the application of the acoustic emission (AE) methodology to composite ceramics under tensile fatigue loading.

The application was carried out on a limitate number of specimens and under loading conditions deeply different among them in terms of frequency (in any case very low), load intensity and loading sequence.

The attention was concentrated almost only to the final part of the tests, just before the failure.

An interesting discussion was developed on the different nature of the AE signals, if in the phase of load increasing (peak) or decreasing (valley). I believe that is an important remark.

Even if the correlation between the AE results and the fracture parameters is well exposed and the results are coherent, the narrow number of tests and its spread in terms of loading parameters don't allow defining general conclusions. In my opinion, the analysis of results and the discussion are too complicated and difficult to read.

Some remarks:

line 56-57: the thermographic techniques were used since the eighties to detect the fatigue limit, the fatigue curve and the damage on specimens and mechanical components. In some case, this technique was coupled with the AE, showing good results in terms of damage analysis.

line 100, 116: Hz.

Table 1: the tests were performed under too different conditions. It is very difficult to define any general rule.

lines 105-107: even if the ER monitoring is shown in another paper, please give some basic information.

lines 120-124: please, give more explanations about the triggering among the acquisition, it is a very important point.

The microstructural analysis try to confirm the AE results, but the authors declare that the specimens have a lot of porosity. The failure mode could depend on this parameter?

It seems that the frequency effect is different for the 0.1 Hz case depending on the load sequence, as well as the anser in terms of AE cumulated energy (Fig. 2 b). Please explain it.

Author Response

Even if the correlation between the AE results and the fracture parameters is well exposed and the results are coherent, the narrow number of tests and its spread in terms of loading parameters don't allow defining general conclusions. In my opinion, the analysis of results and the discussion are too complicated and difficult to read. – THE FOCUS OF THE PAPER IS NOT TO IDENTIFY “THE” FATIGUE FRACTURE MECHANISMS BUT TO SHOW HOW AE CAN INFORM DAMAGE PROGRESSION FOR CASES WHERE DIFFERENT AE BEHAVIOR POINTED TOWARDS DIFFERENT FAILURE PROGRESSION IN FATIGUE TESTS.

Some remarks:

line 56-57: the thermographic techniques were used since the eighties to detect the fatigue limit, the fatigue curve and the damage on specimens and mechanical components. In some case, this technique was coupled with the AE, showing good results in terms of damage analysis. – IN PARTICULAR I AM REFERING TO SIC/SIC WOVEN COMPOSITES WHICH I ADDED TO THE SENTENCE.

line 100, 116: Hz. - CORRECTED

Table 1: the tests were performed under too different conditions. It is very difficult to define any general rule. – I STATED IN THE TEXT THAT THE INITIAL LOAD CHOSEN WAS BASED ON 70% OF THE UTS FOR A GIVEN PANEL. THE INTENT WAS TO PERFORM FATIGUE AT DIFFERENT FREQUENCIES.

lines 105-107: even if the ER monitoring is shown in another paper, please give some basic information. – I REFRAIN FROM DOING THIS SINCE IT IS DISCUSSED IN DETAILS IN THE REFERENCES. THE PAPER ALREADY SEEMS PRETTY LONG. IF YOU REALLY THINK IT NECESSARY THEN I CAN.

lines 120-124: please, give more explanations about the triggering among the acquisition, it is a very important point. – I ADDED IN THE FOLLOWING PARAGRAPH (LINE 141) THE USE OF FIRST ARRIVAL BASED ON WHICH SENSOR IS TRIGGERED FOR A GIVEN EVENT. IS THERE MORE YOU WOULD LIKE EXPLAINED?

The microstructural analysis try to confirm the AE results, but the authors declare that the specimens have a lot of porosity. The failure mode could depend on this parameter? – YES, I ADDED THIS MAY BE THE CASE IN THE THE LAST SENTENCES OF THE DISCUSSION STATES (LINES 500 TO 503)

It seems that the frequency effect is different for the 0.1 Hz case depending on the load sequence, as well as the anser in terms of AE cumulated energy (Fig. 2 b). Please explain it. – I DO NOT HAVE AN EXPLANATION FOR THE TWO DIFFERENT BEHAVIORS OF THE 01 HZ AND I DO NOT SEE WHY THERE WOULD BE A CONNECTION WITH STRESS STEPS. IT IS CLEAR THAT THE LACK OF SIGNIFICANT AE AT THE END OF THE TEST FOR 111-8 IS CONSISTENT WITH THERE NOT BEING ANY VALLEY AE EVENTS.

Round 2

Reviewer 3 Report

The paper was significantly improved in terms of presentation of methods and results.

Even if I believe that the spread of the results is large enough to require a more extensive test campaign, the paper could be published as a first analysis, considering also the correlation performed between the AE data and the microstructural analysis.

Please, correct line 153: ...time what precedes...

Please, correct in figures any notation sec. or seconds with the official one [s]

Author Response

Please, correct line 153: ...time what precedes... - CORRECTED

Please, correct in figures any notation sec. or seconds with the official one [s] - CORRECTED

ALSO MADE SOME MINOR GRAMMAR CORRECTIONS THROUGHOUT AND ADDED A FINAL SENTENCE IN THE CONCLUSIONS:

"The use of the latter AE waveform feature, direction of first extensional peak, was novel for these kinds of tests and may be another useful feature to discern damage mechanisms when using AE for materials and structures testing."